psychology

emotion, empathy, trait mindfulness, faces, speech prosody

**Author for correspondence:**
César F. Lima
e-mail: cesar.lima@iscte-iul.pt

# Higher trait mindfulness is associated with empathy but not with emotion recognition abilities

Ricardo F. Vilaverde, Ana Isabel Correia
and César F. Lima

Centro de Investigação e Intervenção Social (CIS-IUL), Instituto Universitário de Lisboa (ISCTE-IUL), Lisboa, Avenida das Forças Armadas, 1649-026 Portugal

CFL, 0000-0003-3058-7204

Mindfulness involves an intentional and non-judgemental attention or awareness of present-moment experiences. It can be cultivated by meditation practice or present as an inherent disposition or trait. Higher trait mindfulness has been associated with improved emotional skills, but evidence comes primarily from studies on emotion regulation. It remains unclear whether improvements extend to other aspects of emotional processing, namely the ability to recognize emotions in others. In the current study, 107 participants ($M_{age} = 25.48$ years) completed a measure of trait mindfulness, the Five Facet Mindfulness Questionnaire, and two emotion recognition tasks. These tasks required participants to categorize emotions in facial expressions and in speech prosody (modulations of the *tone of voice*). They also completed an empathy questionnaire and attention tasks. We found that higher trait mindfulness was associated positively with cognitive empathy, but not with the ability to recognize emotions. In fact, Bayesian analyses provided substantial evidence for the null hypothesis, both for emotion recognition in faces and in speech. Moreover, no associations were observed between mindfulness and attention performance. These findings suggest that the positive effects of trait mindfulness on emotional processing do not extend to emotion recognition abilities.

## 1. Introduction

Mindfulness can be defined as a state of being attentive to and aware of what is happening in the present moment, in an intentional and non-judgemental way [1]. It involves the self-regulation of attention so that it is maintained on the ongoing experience (e.g. sensations, thoughts). This contrasts with states

of rumination, absorption in the past, anxiety about the future, multitasking or with behaving in a compulsive or automatic way [1]. Research on mindfulness has increased dramatically over the past two decades with the rising popularity of meditation training programmes. A body of work has been examining the benefits of such programmes on physical and mental health, and on attention and emotional processes [2]. Several studies have reported evidence for positive effects of mindfulness interventions in contexts such as chronic pain management, immune function, depression relapse, anxiety and treatment of drug addiction [3].

Current perspectives suggest that mindfulness can also be examined as an inherent disposition or trait, which can arise regardless of experience with formal meditation practice [1,4]. Trait mindfulness refers to the tendency to be mindful in everyday life, and individual differences in this tendency are typically measured with self-report questionnaires [1,5]. There is less research on trait mindfulness than on mindfulness training, but associations have been documented with variables such as self-awareness and self-regulated behaviour, resilience, positive affect, subjective well-being, mental health, life satisfaction, job satisfaction and performance and attention abilities (e.g. [1,6–9]).

Emotions are one of the most investigated topics in trait mindfulness research. According to Brown and Ryan [1], trait mindfulness includes the disposition to be aware of one's emotional states, perceptual clarity about such states, and enhanced attention to and awareness of others' emotional cues during communication. Mindfulness could therefore encourage the development of key abilities related to emotional functioning. This idea is supported by studies on emotion regulation. For example, in an fMRI study, Creswell et al. [10] had participants matching negative facial expressions with appropriate affect labels (angry, scared) or with gender-appropriate names (control task). During affect labelling, individuals scoring higher on trait mindfulness showed greater widespread prefrontal activation and attenuated amygdala responses, as well as a negative relationship between prefrontal and amygdala responses. This suggests enhancements in neural emotion regulation pathways and could potentially explain why mindful individuals can experience reduced negative affect and improved health. Associations between trait mindfulness and improved emotion regulation have been repeatedly reported both in behavioural and neuroscientific work (e.g. [11,12]). Trait mindfulness has also been associated with higher trait emotional intelligence, as indicated by a meta-analysis of 17 studies [13], and with attenuated emotional attentional blink [14]. Negative distractors presented in a rapid visual stream hinder the detection of a subsequent target, but Makowski et al. [14] found that this emotion-induced blindness was shorter in mindful individuals. This suggests a more efficient attentional recovery after emotional distractors and refocusing on the task.

Links between trait mindfulness and emotional processes are well documented, but it remains unclear whether they extend to the ability to recognize emotions in others. Both facial and vocal expressions provide rich information about others' emotional states during social interactions, and emotion recognition skills relate to psychosocial adjustment [15], as well as with lower depression symptoms and relationship well-being [16]. Several variables determine individual differences in emotion recognition, such as cultural background (e.g. [17,18]), musical training [19], age (e.g. [20,21]) or brain pathology (e.g. [22]). As for mindfulness, one study by English et al. [23] tested a sample of 126 female university students and found that those scoring higher on mindfulness needed less perceptual information to recognize facial expressions of fear. The task involved evaluating sequences of faces progressively displaying more emotional content. These findings suggest an association between mindfulness and perceptual aspects of facial processing, at least for fear. Crucially, however, it is unknown whether benefits are also seen for emotion recognition more generally, and whether they are specific to faces or generalize across sensory modalities.

In the present study, participants completed a widely used measure of trait mindfulness, the Five Facet Mindfulness Questionnaire (FFMQ; [5]), and two forced-choice emotion categorization tasks. One was focused on facial expressions, and the other one on speech prosody (modulations of the *tone of voice*). Our primary question was whether trait mindfulness is associated with enhanced emotion recognition across visual and auditory modalities. Participants additionally completed attention and empathy measures. We wanted to explore whether the potential link between mindfulness and emotion recognition is direct, or whether it could be explained by effects of mindfulness on these variables. Trait mindfulness has been associated with more efficient attention networks [7,24,25], and this could plausibly account for enhancements in emotion recognition, because attention has been shown to benefit emotion recognition performance [26,27]. Similarly, trait mindfulness has been associated with higher empathy [28], and empathy could support a better emotion recognition performance ([15,29]; but see [30]).

# 2. Method

## 2.1. Participants

A total of 107 participants took part in this study. They were 25.48 years of age on average (s.d. = 7.68, range = 18–52; 71 women), and had 15.47 years of education (s.d. = 2.92, range = 10–30). Participants were recruited in response to an advertisement (i) posted in local research participant pools, and (ii) sent via email to members of the wider community who were part of the researchers' social networks. Inclusion criteria were age at least 18, no history of neurological or psychiatric disorders, normal hearing, normal or corrected-to-normal vision, and European Portuguese as native language. Most participants had no meditation experience ($n = 70$), but 37 had some experience (number of hours, $M = 183.63$; s.d. = 743.26; range = 1–4380). Ethical approval for the study protocol was obtained from the local Ethics Committee, ISCTE-IUL (reference 31/2018). Written informed consent was collected from all participants.

An *a priori* power analysis conducted with G*Power 3.1 [31] indicated that a sample size of at least 84 was required to detect correlations of $r = 0.30$ or larger between variables, considering an alpha level of 0.05 and a power of 0.80. A sensitivity analysis, using the same alpha and power values, indicated that our actual sample ($N = 107$) had a sensitivity to detect significant correlations of at least $r = 0.27$.

## 2.2. Materials

### 2.2.1. Trait mindfulness

The FFMQ is a 39-item questionnaire grouped into five subscales, each corresponding to a different facet of mindfulness: observing (e.g. *I notice the smells and aromas of things*), describing (e.g. *I'm good at finding words to describe my feelings*), non-judging of inner experience (e.g. *I tell myself I shouldn't be feeling the way I'm feeling*), non-reactivity to inner experience (e.g. *I watch my feelings without getting lost in them*) and acting with awareness (e.g. *I am easily distracted*). Items are rated on a five-point Likert scale from 1 (*never or very rarely true*) to 5 (*very often or always true*). The original FFMQ [5] and the Portuguese translation [32] have sound psychometric properties, including good construct validity and internal consistency. Individual item scores are summed to produce scores for each subscale and a total mindfulness score. Internal consistency values were acceptable-to-excellent in the current dataset (Cronbach's $\alpha = 0.88$ for the full scale, ranging from $\alpha = 0.73$ for non-reactivity to $\alpha = 0.92$ for describing).

### 2.2.2. Empathy

The Questionnaire of Cognitive and Affective Empathy (QCAE) includes 31 items, assessing cognitive empathy (e.g. *I can tell if someone is masking their true emotion*) and affective empathy (e.g. *I am happy when I am with a cheerful group and sad when the others are glum*). Items are rated on a four-point Likert scale from 1 (*strongly disagree*) to 4 (*strongly agree*). The original QCAE [33] and the Portuguese translation [34] have sound psychometric properties, including good construct validity and internal consistency. The Portuguese QCAE has 30 items only (one item was excluded from the original version due to extreme low loading on the respective scale; [34]). Individual item scores are summed to produce cognitive and affective empathy scores, and a total empathy score. Internal consistency values were acceptable-to-good in the current dataset ($\alpha = 0.85$ for the full QCAE questionnaire, $\alpha = 0.79$ for affective empathy and $\alpha = 0.87$ for cognitive empathy).

### 2.2.3. Attention

Attention abilities were measured using the Stroop task and a short version of the Attention Network Test (ANT; [35]). The Stroop task consisted of a series of colour words (red, green, blue or yellow) presented on a computer screen (black background), each of which was displayed in a colour that either matched (congruent) or did not match (incongruent) the meaning of the word. Participants identified the colour in which the words were displayed by pressing the corresponding key on a keyboard. Each trial consisted of a fixation cross for 500 ms, followed by the word presented for 200 ms and a response window of 1700 ms. The inter-trial interval was 1000 ms. Participants completed 144 trials, half of which were incongruent. We calculated the percentage of correct

responses and a Stroop incongruency score (reaction times on incongruent trials – reaction times on congruent trials; including correct trials only).

The short version of the ANT was retrieved from the author's website (http://people.qc.cuny.edu/Faculty/Jin.Fan/Pages/Downloads.aspx). This version consisted of 120 trials divided into 5 runs, preceded by one practice run of 12 trials. For each trial, participants pressed one of two keys, indicating whether a target arrow was pointing left or right. The target arrow was presented above or below a centrally located fixation cross, and it was flanked either by pairs of congruent arrows (congruent condition) or by pairs of incongruent arrows (incongruent condition). Half of the trials were incongruent. Additionally, each trial was preceded either by no cue (40 trials) or by one of two types of cues, consisting of asterisks presented for 100 ms: a centre cue aligned with the fixation cross, indicating that the target arrow was about to show up (40 trials); or a spatial cue, either above or below the fixation cross, indicating where the target arrow would appear (40 trials). The interval between cue and target was 400 ms. Participants were asked to respond as fast and accurately as possible, and the target remained visible until they responded or until 1700 ms after presentation. We calculated the efficiency of attentional networks based on average reaction times, for correct trials only: alerting = no cue – centre cue; orienting = centre cue – spatial cue; and conflict = incongruent target – congruent target [36].

### 2.2.4. Emotion recognition

Participants completed two emotion recognition tasks, one focusing on facial expressions and the other one on speech prosody. Each task included 84 trials, with 12 different stimuli representing each of seven emotions (anger, disgust, fear, happiness, sadness, surprise and neutral). The stimuli were taken from previously validated databases (speech prosody, [37]; facial expressions, Karolinska Directed Emotional Faces database, [38]) and have been used in previous studies (e.g. [19,39–42]). Speech prosody stimuli consisted of short sentences ($M = 1470$ ms, s.d. $= 240$) with emotionally neutral semantic content (e.g. 'O futebol é um desporto', *Football is a sport*), produced by two female speakers to communicate emotions with prosodic cues alone (i.e. variations in pitch, loudness, timing and voice quality). Facial expressions consisted of colour photographs of male and female actors with no beards, moustaches, earrings, eyeglasses or visible make-up. Each photograph was presented for 2000 ms. The two tasks were similarly difficult (based on validation data, average recognition accuracy was 75.60% for speech prosody and 79.43% for facial expressions).

Participants made an eight-alternative forced-choice judgement for each stimulus, selecting the emotion that was being expressed from a list including *neutrality, anger, disgust, fear, happiness, sadness, surprise* and *none of the above*. Each of the tasks started with four practice trials. The 84 experimental trials that followed were randomized for each participant. Each stimulus was presented once (after a 1000 ms fixation cross) and no feedback was given.

Accuracy rates were calculated for each emotion and task, and the analyses that follow were based on average scores for each task. Internal consistency values were acceptable-to-good for both tasks: $\alpha = 0.75$ for facial expressions, and $\alpha = 0.85$ for speech prosody. The accuracy data were arcsine square-root transformed and corrected for possible response biases using unbiased hit rates, or *Hu* ([43]; for a discussion of biases in forced-choice tasks, e.g. [44]). Hu values represent the joint probability that a given emotion will be correctly recognized (given that it is presented), and that a given response category will be correctly used (given that it is used at all), such that they vary between 0 and 1. Hu = 0 when no stimulus from a given emotion is correctly recognized, and Hu = 1 only when all the stimuli from a given emotion (e.g. happy prosody) are correctly recognized, and the corresponding response category (e.g. happiness) is always correctly used (i.e. when there are no false alarms). The response category 'none of the above' was rarely selected (5.46% on average across modalities).

## 2.3. Procedure

Participants were tested in small groups (up to four participants) in a quiet room. They completed the emotion recognition tasks, the attention tasks and then the demographic, mindfulness and empathy questionnaires. The order of the emotion recognition and attention tasks was counterbalanced across participants. The testing session lasted about 1 hour, and short breaks were allowed between tasks. The auditory stimuli were presented via high-quality headphones, with the volume adjusted to a comfortable level for each participant. The attention and emotion recognition tasks were implemented

**Table 1.** Descriptive statistics for the questionnaires and for the attention and emotion recognition tasks.

| | M | s.d. | range |
|---|---|---|---|
| FFMQ (total) | 122.37 | 16.73 | 86–165 |
| Observing | 27.46 | 5.22 | 13–38 |
| Describing | 25.91 | 6.21 | 12–40 |
| Acting with Awareness | 23.79 | 5.35 | 10–33 |
| Non-judging | 24.20 | 6.43 | 8–40 |
| Non-reactivity | 21.03 | 4.00 | 8–34 |
| QCAE (total) | 94.36 | 9.78 | 67–116 |
| Affective Empathy | 34.05 | 5.00 | 20–43 |
| Cognitive Empathy | 60.32 | 7.58 | 40–75 |
| Stroop | | | |
| Accuracy | 0.93 | 0.08 | 0.51–1.00 |
| Incongruency Score | 67.21 | 42.04 | −31.05–226.46 |
| Attention Network Test | | | |
| Alerting | 26.11 | 24.25 | −34.85–115.65 |
| Orienting | 50.27 | 30.66 | −77.73–122.50 |
| Executive Control | 86.51 | 30.28 | 8.03–157.75 |
| Emotion Recognition (average, Hu scores) | 0.65 | 0.11 | 0.21–0.83 |
| Faces | 0.68 | 0.11 | 0.28–0.88 |
| Prosody | 0.62 | 0.15 | 0.14–0.87 |

in E-Prime 2.0 (version 2.0.10.356). Due to software malfunction, data from two participants on the ANT were not recorded. These participants were therefore excluded from all analyses including this variable.

## 2.4. Statistical analysis

The data were statistically evaluated based on standard frequentist *and* Bayesian approaches. In each analysis, a Bayes factor ($BF_{10}$) statistic was estimated, which considers the likelihood of the observed data given the alternative and null hypotheses. These analyses were conducted on JASP Version 0.10.2 [45], using the default priors (correlations, stretched beta prior width = 1). $BF_{10}$ values were interpreted following Jeffreys' guidelines [46], such that values between 1 and 3 correspond to anecdotal evidence for the alternative hypothesis, between 3 and 10 to substantial evidence, between 10 and 30 to strong evidence, between 30 and 100 to very strong evidence and greater than 100 to decisive evidence. A $BF_{10}$ less than 1 corresponds to evidence in favour of the null hypothesis: values between 0.33 and 1 correspond to anecdotal evidence, between 0.10 and 0.33 to substantial evidence, between 0.03 and 0.10 to strong evidence, between 0.01 and 0.03 to very strong evidence and less than 0.01 to decisive evidence. Thus, one important advantage of Bayesian statistics over the frequentist approach is that they allow us to interpret null results and to formally draw inferences based on them.

# 3. Results

Table 1 shows descriptive statistics for the questionnaires and for the attention and emotion recognition tasks (in electronic supplementary material, table S1 shows statistics for each emotion on the emotion recognition tasks). The average scores and subscores on the FFMQ and QCAE are consistent with previous results [34,47]. Also in line with previous studies, there were small to medium positive correlations among FFMQ subscales, ranging from $r = 0.22$ to $r = 0.45$ (electronic supplementary material, table S2). The correlation between cognitive and affective empathy did not reach significance, $r = 0.17$, $p = 0.08$, $BF_{10} = 0.57$.

**Table 2.** Pairwise correlations between mindfulness (FFMQ total) and other variables.

| variable | *r* | *p*-value | $BF_{10}$ |
|---|---|---|---|
| QCAE (total) | 0.21 | 0.03 | 1.34 |
| Affective Empathy | −0.10 | 0.33 | 0.19 |
| Cognitive Empathy | 0.34 | <0.001 | 61.98 |
| Stroop | | | |
| Accuracy | 0.05 | 0.64 | 0.14 |
| Incongruency Score | 0.03 | 0.75 | 0.13 |
| Attention Network Test | | | |
| Alerting | 0.00 | 0.98 | 0.12 |
| Orienting | −0.05 | 0.62 | 0.14 |
| Executive Control | −0.10 | 0.30 | 0.21 |
| Emotion Recognition (average, Hu scores) | −0.01 | 0.30 | 0.20 |
| Faces | −0.10 | 0.31 | 0.20 |
| Prosody | −0.07 | 0.50 | 0.15 |

Participants performed generally well on the emotion recognition tasks, both for facial and prosodic expressions (table 1). Average accuracy rates were above the chance level (approx. 0.14), and there was a significant positive correlation across the two tasks, $r = 0.35$, $p < 0.001$, $BF_{10} = 83.09$. Despite the generally high performance, there was wide variability across participants, with scores ranging from approximately 0.20 to 0.88 (s.d. $\cong 0.13$; table 1). This is comparable to previous studies on the correlates of individual differences in emotion recognition (e.g. [48–50]).

Table 2 shows zero-order correlations between mindfulness and the remaining study variables. Contrary to our hypothesis, we found no associations between FFMQ scores and emotion recognition, strongest $r = -0.10$, lowest $p = 0.30$. In fact, Bayesian statistics provided substantial evidence for the null hypothesis, both for facial and prosodic expressions, highest $BF_{10} = 0.20$. Follow-up analyses revealed that null results were also observed for FFMQ subscores, strongest $r = -0.10$, lowest $p = 0.14$, highest $BF_{10} = 0.35$, and when the focus was on specific emotions rather than on average emotion recognition (see electronic supplementary material, table S3 for details). We found some significant negative associations for happiness (with FFMQ total scores and Describing subscores) and surprise (with Non-judging subscores), but these were small, strongest $r = -0.22$, and would not survive Bonferroni corrections for multiple comparisons, highest $BF_{10} = 1.49$.

To exclude the possibility that null results were due to the effects of demographic variables, we used multiple regression, modelling average accuracy on emotion recognition as a function of FFMQ scores, age, sex and education. None of the predictor variables made an independent contribution to the model, lowest $p = 0.34$, highest $BF_{10} = 0.59$, and the model itself was not significant, as indicated by both frequentist and Bayesian statistics, $R = 0.15$, $F_{4,102} = 0.58$, $p = 0.68$, $BF_{10} = 0.02$. The same was obtained for similar models conducted for each emotion recognition task separately, and for models including the five FFMQ subscores as predictor variables. Thirty-seven participants had some meditation experience, and the number of hours of experience correlated with FFMQ scores, $r = 0.30$, $p = 0.002$, $BF_{10} = 13.42$. However, in a regression model including FFMQ scores and hours of experience as predictor variables, none of them contributed independently to explain variance in emotion recognition, lowest $p = 0.41$, highest $BF_{10} = 0.41$, and the model was not significant, $R = 0.09$, $F_{2,102} = 0.43$, $p = 0.66$, $BF_{10} = 0.09$.

We finally focused on how mindfulness and emotion recognition related to empathy and attention (table 3). Emotion recognition was not correlated with empathy, strongest $r = 0.08$, lowest $p = 0.41$, highest $BF_{10} = 0.17$. It was also not correlated with attention, except for an unpredicted association between facial emotion recognition and ANT orienting scores, $r = 0.29$, $p = 0.003$, $BF_{10} = 11.37$. Trait mindfulness was not correlated with attention, strongest $r = -0.19$, lowest $p = 0.06$, highest $BF_{10} = 0.70$ (table 2 and table 4 for details), but there were associations with empathy. We found a significant association between FFMQ scores and higher cognitive empathy, $r = 0.34$, $p < 0.001$, and Bayesian

**Table 3.** Correlations between emotion recognition accuracy and empathy and attention. $^*p < 0.05$; $BF_{10}$ values are indicated in parenthesis.

| | emotion recognition (Hu scores) | | |
| --- | --- | --- | --- |
| | average | faces | prosody |
| QCAE (total) | 0.07 (0.07) | 0.02 (0.12) | 0.08 (0.17) |
| Affective Empathy | 0.04 (0.04) | 0.02 (0.12) | 0.05 (0.14) |
| Cognitive Empathy | 0.06 (0.14) | 0.01 (0.12) | 0.07 (0.16) |
| Stroop | | | |
| Accuracy | 0.13 (0.30) | 0.05 (0.14) | 0.16 (0.46) |
| Incongruency Score | −0.08 (0.17) | −0.20* (0.91) | 0.03 (0.13) |
| Attention Network Test | | | |
| Alerting | 0.17 (0.49) | 0.13 (0.30) | 0.14 (0.31) |
| Orienting | 0.18 (0.62) | 0.29* (11.37) | 0.03 (0.13) |
| Executive Control | −0.19 (0.71) | −0.13 (0.30) | −0.17 (0.54) |

**Table 4.** Correlations between FFMQ subscales and empathy and attention. $^*p < 0.05$; $^{**}p < 0.01$; $BF_{10}$ values are indicated in parenthesis.

| | FFMQ subscales | | | | |
| --- | --- | --- | --- | --- | --- |
| | observing | describing | acting with awareness | non-judging | non-reactivity |
| QCAE (total) | 0.20* (1.06) | 0.19 (0.79) | 0.21* (1.19) | −0.06 (0.14) | 0.15 (0.39) |
| Affective Empathy | 0.09 (0.18) | −0.03 (0.13) | 0.07 (0.16) | −0.18 (0.71) | −0.27** (5.71) |
| Cognitive Empathy | 0.20* (1.07) | 0.26** (4.57) | 0.22* (1.65) | 0.05 (0.14) | 0.37** (>100) |
| Stroop | | | | | |
| Accuracy | 0.06 (0.15) | 0.00 (0.12) | 0−0.01 (0.12) | 0.01 (0.12) | 0.12 (0.24) |
| Incongruency Score | −0.00 (0.12) | 0.02 (0.12) | 0.16 (0.43) | −0.07 (0.16) | 0.01 (0.12) |
| Attention Network Test | | | | | |
| Alerting | 0.05 (0.14) | −0.02 (0.12) | 0.06 (0.15) | −0.04 (0.13) | −0.04 (0.13) |
| Orienting | 0.11 (0.23) | −0.03 (0.13) | −0.18 (0.70) | −0.06 (0.14) | 0.03 (0.13) |
| Executive Control | −0.16 (0.43) | −0.07 (0.16) | −0.00 (0.12) | 0.02 (0.12) | −0.13 (0.30) |

statistics indicated that the evidence was very strong, $BF_{10} = 61.98$. Follow-up analyses focusing on the FFMQ subscores showed that correlations are evident for Non-reactivity, Describing, Acting with Awareness, and Observing, weakest $r = 0.20$, highest $p = 0.04$, lowest $BF_{10} = 1.07$ (table 4 for details). No associations were found with affective empathy, apart from a negative correlation with Non-reactivity scores, $r = −0.27$, $p = 0.01$, $BF_{10} = 5.71$.

# 4. Discussion

In the present study, we found no evidence that trait mindfulness is associated with the ability to recognize emotions in others. For facial expressions and speech prosody, emotion recognition performance was *similar* in participants scoring higher or lower on mindfulness, as indicated by frequentist and Bayesian statistics. Trait mindfulness has been associated with benefits in emotional variables, in studies focused on emotion regulation (e.g. [10–12]) or on subjective measures of felt

emotional states (e.g. [6,51]). Our results suggest that the benefits do not extend to emotion recognition abilities. This seems unexpected, considering that trait mindfulness has been associated with emotional intelligence [13], and the ability to recognize emotions is one of the components of this construct. The strength of the association between mindfulness and emotional intelligence varies widely across studies, however [13]. Most of these studies additionally rely on self-reported emotional intelligence, and not on performance-based measures, which would be more comparable to the ones used here.

Our results also seem to contrast with those by English *et al.* [23] pointing to a link between mindfulness and facial emotional processing. They used the same mindfulness questionnaire as in the current study (FFMQ), and the internal consistency of their self-report and performance measures is comparable to ours. The discrepancy in results might stem from differences in samples, or the particular way emotion recognition was assessed. English *et al.* [23] asked participants to recognize emotions in 10-image sequences starting with a neutral expression and ending with a full expression. Their focus was on the *amount* of information needed for recognition, not on the ability to recognize emotions. Moreover, their results were seen for fearful expressions only, as assessed exclusively by female participants. The potential role of task in associations between mindfulness and emotion recognition will need to be addressed in future studies. The benefits could be subtle and more likely to be seen in challenging conditions, which our tasks did not allow us to assess. This could be studied by systematically manipulating stimulus ambiguity or cognitive load (e.g. [23,48]). It will also be important to expand the current results, by combining measures of trait mindfulness like the one used here with other measures of mindfulness that do not rely on self-report. This could be done using, for example, experience-sampling methods or performance-based tasks of state mindfulness (e.g. [1]).

Another null finding of the current study was that trait mindfulness had no association with attention abilities. It is often assumed that mindfulness and attention are closely related because training of attention skills is central in meditation practices (e.g. [52,53]). In fact, there is evidence for enhanced attentional processes in mindfulness practitioners (e.g. [2,53]). When it comes to trait mindfulness, however, evidence is much weaker. Some studies suggest that higher trait mindfulness relates to more efficient attentional networks (e.g. [24,54]), but such advantages are not always replicable, even in well-powered studies using reliable measures [55]. Advantages are also often limited to one or two of multiple attention measures and facets of mindfulness [25,56]. Our results add to the growing evidence for a weak or non-existent relationship between trait mindfulness and attention. It is possible that attention does not share the same relationship with mindfulness as a practice and as a trait [55], but this warrants further investigation.

Results for emotion recognition and attention were null, but we found robust evidence that trait mindfulness is associated with empathy. This was most evident for cognitive empathy, and it reflects a general result observed across most facets of the FFMQ. These findings corroborate those by Greason and Cashwell [57] with a different empathy questionnaire (Interpersonal Reactivity Index), and those by MacDonald & Price [28] with the same questionnaire that we used here (QCAE). MacDonald & Price [28] reported a pattern of results similar to ours: small-to-medium positive correlations between trait mindfulness and cognitive empathy, and non-significant or negative correlations with affective empathy. An intentional attention to one's feelings and thoughts is a core feature of mindfulness [1], and this could facilitate an understanding of others' thoughts, cognitions and emotions, which is central for empathic responding. What remains to explain is why mindfulness relates to cognitive and affective empathy in a different manner. It could be because affective empathy implies being emotionally affected by others' experiences or problems, and trait mindfulness encourages the opposite. For example, the Non-reactivity facet broadly assesses the ability *not* to react to feelings, thoughts or emotions, even if they are difficult [5]. Consistent with this, Non-reactivity was the facet showing the strongest positive association with cognitive empathy, and negative association with affective empathy.

Empathy was associated with trait mindfulness, but not with emotion recognition. Bayesian analyses actually provided substantial-to-strong evidence for the null hypothesis. Besel & Yuille [29] had previously shown an association between empathy and higher facial emotion recognition. However, this was found for fearful expressions only, and when participants were exposed to the stimuli for a long time. In other words, it could not be generalized across emotions and task conditions. More recently, Olderbak & Wilhelm [30] conducted four studies with different designs and sample characteristics, and found that the relationship between empathy and facial emotion recognition is either not significant or weak. As for emotional prosody, to our knowledge no previous studies have

documented associations with empathy, and studies focusing on potential correlations with personality traits have reported null results [58]. It could be that emotion recognition and empathy reflect dissociable processes, with emotion recognition depending relatively more on low-level perceptual processes, and empathy on higher order affective and cognitive mechanisms (e.g. [59]). According to Olderbak & Wilhelm [30], it could also be that the lack of an association reflects differences in the measurement approaches. While empathy is assessed as self-reported typical behaviour (i.e. a personality-like construct), emotion recognition is assessed as maximal effort using performance-based tasks.

To conclude, the current study showed that trait mindfulness is not associated with the ability to recognize emotions in facial or prosodic expressions. We also found no associations between mindfulness and attention, but documented a robust link with cognitive empathy. Our results add to the growing literature on the correlates of trait mindfulness, thus contributing to a better understanding of this construct, and its role in positive psychological experience and skills (e.g. [1,60]). They emphasize the notion that the different aspects of emotional processing might relate to mindfulness in different ways. While trait mindfulness might serve important self-regulatory and awareness functions, its role for more low-level emotional-perceptual mechanisms might be less apparent. These findings might additionally have broader implications for how mindfulness is used in clinical practice (e.g. for exploring the facets, potential benefits and limits of mindfulness with clients).

Research ethics. Ethical approval for the study protocol was obtained from the local Ethics Committee, ISCTE-IUL (reference 31/2018). Written informed consent was collected from all participants.

Data accessibility. The dataset supporting this article is available as electronic supplementary material. The dataset is also freely available for public use at Dryad: https://doi.org/10.5061/dryad.2rbnzs7j2 [61].

Authors' contributions. R.F.V. prepared the tasks, collected the data, participated in data analysis and interpretation, participated in the design of the study and drafted the manuscript. A.I.C. helped preparing the experimental tasks and participated in data analysis and interpretation. C.F.L. conceived and designed the study, coordinated the study, participated in data analysis and interpretation, and helped draft the manuscript. All authors gave final approval for publication.

Competing interests. César F. Lima was a member of the Royal Society Open Science editorial board at the time of submission; however, they were not involved in the editorial assessment of the manuscript in any way.

Funding. Funded by a grant from Portuguese Foundation for Science and Technology (FCT) awarded to C.F.L. (IF/00172/2015).

Acknowledgements. We thank Aissa Baldé for helping with data collection.

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
