## [Reviewer comments · Royal Society Open Science]

Review History

RSOS-192077.R0 (Original submission)

Review form: Reviewer 1

Is the manuscript scientifically sound in its present form?

No

Are the interpretations and conclusions justified by the results?

Yes

Is the language acceptable?

Yes

Do you have any ethical concerns with this paper?

No

Have you any concerns about statistical analyses in this paper?

No

Recommendation?

Major revision is needed (please make suggestions in comments)

Comments to the Author(s)

Please see attached file (Appendix A).

Review form: Reviewer 2**Is the manuscript scientifically sound in its present form?**

Yes

Are the interpretations and conclusions justified by the results?

Yes

Is the language acceptable?

Yes

Do you have any ethical concerns with this paper?

No

Have you any concerns about statistical analyses in this paper?

No

Recommendation?

Accept with minor revision (please list in comments)

Comments to the Author(s)

This manuscript investigates the relation between mindfulness and the ability to recognize emotions expressed in the voice or face. I appreciate the comprehensive presentation of results, through the numerous tables, and that the data have been made open access. I also don't find the null results surprising, and in the area of socio-emotional research, I think null results are very informative.

The following are my concerns with this manuscript and/or suggestions for its improvement.

On page 4, please specify whether you are discussing trait emotional intelligence or ability emotional intelligence.

In the introduction, but also for the lack of a relation between emotion perception and empathy, you might want to review the manuscript by Olderbak & Wilhelm (Emotion, 2017). They discuss how differences in measurement design, namely typical behavior versus maximal effort, lowers correlations. That paper, because it presents results from 4 studies, is also a notable contrast to those who have found a positive relation between empathy and emotion perception.

Please provide more information about the sample. How were participants recruited? Was there inclusion criteria? Were there any dropouts or missing data? If so, how was that handled?

With the power analysis, please calculate what effect size could be identified, with the present sample and your power requirement.

Please include Cronbach's alpha for all measures, including all FFMQ subscales, and the ability tests.

What was the name of the emotion perception tests?

Please explain why the bayesian analyses were conducted and what they add beyond the frequentist results.

Please provide a brief interpretation of the results, that accomodates the p value and BF10 value.

At the bottom of page 11, shouldnt it be BFs10 instead of BF10? Or is the BFs10 earlier a typo?

A path analysis in confirmatory factor analysis could be a better way to model the relations between mindfulness, empathy, attention, and emotion perception. That would also allow you to model latent variables, which allows one to relate constructs, instead of measures, thus controlling for measurement error.

Finally, in the discussion, comparing your null results with the significant findings of others, I wonder also about the statistical power in the other samples. Are your effect sizes comparable to what others have found? Were their measures more precise (i.e., higher internal consistency)?

Decision letter (RSOS-192077.R0)

09-Mar-2020

Dear Dr Lima,

The editors assigned to your paper ("Higher Trait Mindfulness is Associated with Empathy But Not With Emotion Recognition Abilities") have now received comments from reviewers. We would like you to revise your paper in accordance with the referee and Associate Editor suggestions which can be found below (not including confidential reports to the Editor). Please note this decision does not guarantee eventual acceptance.

Please submit a copy of your revised paper before 01-Apr-2020. Please note that the revision deadline will expire at 00.00am on this date. If we do not hear from you within this time then it will be assumed that the paper has been withdrawn. In exceptional circumstances, extensions may be possible if agreed with the Editorial Office in advance. We do not allow multiple rounds of revision so we urge you to make every effort to fully address all of the comments at this stage. If deemed necessary by the Editors, your manuscript will be sent back to one or more of the original reviewers for assessment. If the original reviewers are not available, we may invite new reviewers.

- Data accessibility

<http://datadryad.org/submit?journalID=RSOS&manu=RSOS-192077>

- Competing interests

- Authors' contributions

- Acknowledgements

- Funding statement

Best regards,
Lianne Parkhouse
Editorial Coordinator

on behalf of Dr Antonia Hamilton (Associate Editor) and Essi Viding (Subject Editor)
openscience@royalsociety.org

Reviewers' Comments to Author:

Reviewer: 1
Comments to the Author(s)

Please see attached file

Reviewer: 2
Comments to the Author(s)

This manuscript investigates the relation between mindfulness and the ability to recognize emotions expressed in the voice or face. I appreciate the comprehensive presentation of results, through the numerous tables, and that the data have been made open access. I also don't find the null results surprising, and in the area of socio-emotional research, I think null results are very informative.

The following are my concerns with this manuscript and/or suggestions for its improvement.

On page 4, please specify whether you are discussing trait emotional intelligence or ability emotional intelligence.

In the introduction, but also for the lack of a relation between emotion perception and empathy, you might want to review the manuscript by Olderbak & Wilhelm (Emotion, 2017). They discuss how differences in measurement design, namely typical behavior versus maximal effort, lowers correlations. That paper, because it presents results from 4 studies, is also a notable contrast to those who have found a positive relation between empathy and emotion perception.

Please provide more information about the sample. How were participants recruited? Was there inclusion criteria? Were there any dropouts or missing data? If so, how was that handled?

With the power analysis, please calculate what effect size could be identified, with the present sample and your power requirement.

Please include Cronbach's alpha for all measures, including all FFMQ subscales, and the ability tests.

What was the name of the emotion perception tests?

Please explain why the Bayesian analyses were conducted and what they add beyond the frequentist results.

Please provide a brief interpretation of the results, that accommodates the p value and BF₁₀ value.

At the bottom of page 11, shouldn't it be BF_{s10} instead of BF₁₀? Or is the BF_{s10} earlier a typo?

A path analysis in confirmatory factor analysis could be a better way to model the relations between mindfulness, empathy, attention, and emotion perception. That would also allow you to model latent variables, which allows one to relate constructs, instead of measures, thus controlling for measurement error.

Finally, in the discussion, comparing your null results with the significant findings of others, I wonder also about the statistical power in the other samples. Are your effect sizes comparable to what others have found? Were their measures more precise (i.e., higher internal consistency)?

Author's Response to Decision Letter for (RSOS-192077.R0)

See Appendix B.

RSOS-192077.R1 (Revision)

Review form: Reviewer 1

Is the manuscript scientifically sound in its present form?

Yes

Are the interpretations and conclusions justified by the results?

Yes

Is the language acceptable?

Yes

Do you have any ethical concerns with this paper?

No

Have you any concerns about statistical analyses in this paper?

No

Recommendation?

Accept as is

Comments to the Author(s)

Thank you for addressing the reviewers' comments and critiques. I believe that the paper is now appropriate for publication.

Decision letter (RSOS-192077.R1)

Dear Dr Lima,

It is a pleasure to accept your manuscript entitled "Higher Trait Mindfulness is Associated with Empathy But Not With Emotion Recognition Abilities" in its current form for publication in Royal Society Open Science. The comments of the reviewer(s) who reviewed your manuscript are included at the foot of this letter.

on behalf of Dr Antonia Hamilton (Associate Editor) and Essi Viding (Subject Editor)
openscience@royalsociety.org

Reviewer comments to Author:
Reviewer: 1

Comments to the Author(s)
Thank you for addressing the reviewers' comments and critiques. I believe that the paper is now appropriate for publication.

Appendix A

This study examined associations between dispositional mindfulness and emotion recognition in a sample of 107 adult participants. The researchers investigated associations between five facets of mindfulness (i.e., observing, describing, acting with awareness, nonjudging, and nonreacting) and two dimensions of emotion recognition, specifically, facial expressions and speech prosody. Using Bayesian statistics, the authors found evidence supporting the null hypothesis for associations between mindfulness and each domain of emotion recognition. This paper contributes to the field of mindfulness by examining its associations with performance-based measurements of emotion recognition, however the paper would benefit from a clearer conceptual model and greater elaboration in places.

Introduction

The Introduction would be strengthened by a more clearly defined conceptual framework and improved organization. The first paragraph of this section would benefit from reorganization, as it does not seem to have a cohesive focus. For example, it's unclear why the authors discuss the MAAS or the FFMQ at this point in the introduction or how this relates to the current study (besides the fact that the authors use the FFMQ in their study). Further, throughout the Introduction, greater justification for examining attention in this study should be explained. It's unclear why this study, which seems to have a central focus on understanding the relationship between mindfulness and emotion recognition, has also included a performance-based measure of attention.

The second paragraph of the Introduction is also somewhat unclear. The authors initially describe the important relationship between emotions and mindfulness. However, this reviewer had difficulty then following the relevance of the Creswell et al. (2007) fMRI research to the current study. Research summarized later in the paragraph supporting associations between

mindfulness and emotion regulation seems importantly connected to the current research question, but the Creswell et al., 2007 findings, with implications for mindfulness intervention impacting negative affect, does not seem directly related to the current study. The end of this paragraph (p. 4) discusses associations between attention and mindfulness, and this section would benefit from a clearer connection between attention and this study's research question and conceptual model.

Method

The authors should consider presenting information on how participants were recruited.

The Materials section would benefit from the presentation of Cronbach's alphas for each of the measures and subscales used in this study.

On page 8, the authors write, "Participants completed two identical emotion recognition tasks, one focusing on facial expressions and the other one on speech prosody." It's unclear how these two measures are "identical." Please clarify.

Can the authors provide further rationale for the statistical approach used? Further, what statistical package was used? What data cleaning procedures were followed, if any?

Additionally, in the statistical analysis section, it would help to provide a clearer description of the analytic plan.

Results

Although values are included in the Tables, it would be helpful to the reader if the authors presented statistics, including r and p values, in the last paragraph of the Results section (p. 11-12).

Discussion

This section of the paper was formatted in a different font from the rest of the paper. Please keep font consistent throughout the manuscript.

It is surprising that emotion recognition was not associated with empathy, as emotion recognition is widely recognized as one of the components of emotion regulation. Can the authors speculate further on why this study may not have found such an association?

Despite presenting power analyses indicating that this study's sample size is sufficient to detect significant differences between variables, is it possible that sample size limitations contributed to the null findings? Is it possible that there was insufficient variability in the emotion recognition measures to detect differences across levels of mindfulness?

What are the limitations of this study?

This paper would benefit from greater attention in both the Introduction and Discussion to the importance and implications of the authors' study question, as well as the findings of this study.

Are there clinical implications for these findings, for example?

Appendix B

Response to the comments

Reviewer 1

1. This study examined associations between dispositional mindfulness and emotion recognition in a sample of 107 adult participants. The researchers investigated associations between five facets of mindfulness (i.e., observing, describing, acting with awareness, nonjudging, and nonreacting) and two dimensions of emotion recognition, specifically, facial expressions and speech prosody. Using Bayesian statistics, the authors found evidence supporting the null hypothesis for associations between mindfulness and each domain of emotion recognition. This paper contributes to the field of mindfulness by examining its associations with performance-based measurements of emotion recognition, however the paper would benefit from a clearer conceptual model and greater elaboration in places.

We thank the Reviewer for their careful reading of our work and for the constructive suggestions. We respond to each specific concern below.

The Introduction would be strengthened by a more clearly defined conceptual framework and improved organization. The first paragraph of this section would benefit from reorganization, as it does not seem to have a cohesive focus. For example, it's unclear why the authors discuss the MAAS or the FFMQ at this point in the introduction or how this relates to the current study (besides the fact that the authors use the FFMQ in their study). Further, throughout the Introduction, greater justification for examining attention in this study should be explained. It's unclear why this study, which seems to have a central focus on understanding the relationship between mindfulness and emotion recognition, has also included a performance-based measure of attention.

We have revised the Introduction following the Reviewer's suggestions.

References to the MAAS and FFMQ have been removed from the first paragraph, and the last paragraph has been substantially re-written to better explain why we examined attention (p. 5-6):

'In the present study, participants completed a widely used measure of trait mindfulness, the Five Facet Mindfulness Questionnaire (FFMQ; Baer et al., 2006), and two forced-choice emotion categorization tasks. One was focussed on facial expressions, and the other one on speech prosody (modulations of the *tone of voice*). Our primary question was whether trait mindfulness is associated with enhanced emotion recognition across visual and auditory modalities. Participants additionally completed attention and empathy measures. We wanted to explore whether the potential link between mindfulness and emotion recognition is direct, or whether it could be explained by effects of mindfulness on these variables. Trait mindfulness has been associated with more efficient attention networks (Di Francesco et al., 2017; Riggs, Black, & Ritt-Olson, 2015; Sørensen et al., 2018), and this could plausibly account for enhancements in emotion recognition, because attention has been shown to benefit emotion recognition performance (Lima, Anikin, Monteiro, Scott, & Castro, 2019; Tsotsi, Bozikas, & Kosmidis, 2015). Similarly, trait mindfulness has been associated with higher empathy (e.g., MacDonald & Price, 2017), and empathy could support a better emotion recognition performance (Besel & Yuille, 2010; Hall, Andrzejewski, & Yopchick, 2009; but see Olderbak & Wilhelm, 2017).'

2. The second paragraph of the Introduction is also somewhat unclear. The authors initially describe the important relationship between emotions and mindfulness. However, this reviewer had difficulty then following the relevance of the Creswell et al. (2007) fMRI research to the current study. Research summarized later in the paragraph supporting associations between mindfulness and emotion regulation seems importantly connected to the current research question, but the Creswell et al., 2007 findings, with implications for mindfulness intervention impacting negative affect, does not seem directly related to the current study. The end of this paragraph (p. 4) discusses associations between

attention and mindfulness, and this section would benefit from a clearer connection between attention and this study's research question and conceptual model.

Our description of the Creswell et al. (2007) study was indeed unclear. Creswell et al. focus on the relation between *trait mindfulness* and emotion regulation, and not on mindfulness interventions, as our previous phrasing suggested. We have revised the text for clarity (p. 4):

'For example, in an fMRI study, Creswell, Way, Eisenberger, and Lieberman (2007) had participants matching negative facial expressions with appropriate affect labels ('angry', 'scared') or with gender-appropriate names (control task). During affect labelling, individuals scoring higher on trait mindfulness showed greater widespread prefrontal activation and attenuated amygdala responses, as well as a negative relationship between prefrontal and amygdala responses. This suggests enhancements in neural emotion regulation pathways, and could potentially explain why mindful individuals can experience reduced negative affect and improved health.'

We refrained from making a connection between attention and the current study at the end of this paragraph, because the Makowsky et al. (2019) findings are about emotional distractors, not about attention abilities in general, as in our study. We therefore decided to leave the topic of attention abilities to the last paragraph for readability (p. 6).

3. Method

The authors should consider presenting information on how participants were recruited.

This information has been added (p. 6):

'Participants were recruited in response to an advertisement (1) posted in local research participant pools, and (2) sent via-email to members of the wider community who were part of the researchers' social networks.'

4. The Materials section would benefit from the presentation of Cronbach's alphas for each of the measures and subscales used in this study.

We agree. This has been added (p. 7, 8 and 10).

FFMQ:

'Internal consistency values were acceptable-to-excellent in the current dataset (Cronbach's $\alpha = .88$ for the full scale, ranging from $\alpha = .73$ for nonreactivity to $\alpha = .92$ for describing).'

QCAE:

'Internal consistency values were acceptable-to-good in the current dataset ($\alpha = .85$ for the full QCAE questionnaire, $\alpha = .79$ for affective empathy, and $\alpha = .87$ for cognitive empathy).'

Emotion recognition:

'Internal consistency values were acceptable-to-good for both tasks: $\alpha = .75$ for facial expressions, and $\alpha = .85$ for speech prosody.'

5. On page 8, the authors write, "Participants completed two identical emotion recognition tasks, one focusing on facial expressions and the other one on speech prosody." It's unclear how these two measures are "identical." Please clarify.

We agree that the word 'identical' is confusing here and have removed it.

6. Can the authors provide further rationale for the statistical approach used? Further, what statistical package was used? What data cleaning procedures were followed, if any? Additionally, in the statistical analysis section, it would help to provide a clearer description of the analytic plan.

We have used JASP Version 0.10.2, as indicated on p. 11.

The main reason for combining standard frequentist with Bayesian statistics is that with Bayesian statistics we could quantify the evidence for the null hypothesis. We could formally draw inferences based on null results, which would be difficult based on p values. This was added on p. 11:

'Thus, one important advantage of Bayesian statistics over the standard approach is that they allow us to interpret null results and to formally draw inferences based on them.'

As for data cleaning procedures, no data points were excluded from the analyses. We have made corrections to the raw emotion recognition data, however, as explained in detail on p. 10.

Regarding the analytic plan, our analyses are generally simple and standard (descriptive statistics, correlations, multiple regressions), and we thought it would be clearer to mention them as the results are described. We refrained from including these details in the Statistical Analysis section to avoid repetition, but we are happy to reconsider in case we misunderstood the Reviewer's suggestion.

7. Results

Although values are included in the Tables, it would be helpful to the reader if the authors presented statistics, including r and p values, in the last paragraph of the Results section (p. 11- 12).

We agree. The paragraph now reads as follows (p. 13):

'We finally focussed on how mindfulness and emotion recognition related to empathy and attention (see Table 3). Emotion recognition was not correlated with empathy, strongest $r = .08$, lowest $p = .41$, highest $BF_{10} = 0.17$. It was also not correlated with attention, except for an unpredicted association between facial emotion recognition and ANT orienting scores, $r = .29$, $p = .003$, $BF_{10} = 11.37$. Mindfulness was not correlated with attention, strongest $r = -.19$, lowest $p = .06$, highest $BF_{10} = 0.70$ (see Table 2 and Table 4 for details), but there were associations with empathy. We found very strong evidence for an association between FFMQ scores and higher cognitive empathy, $r = .34$, $p < .001$, $BF_{10} = 61.98$. Follow-up analyses focussing on the FFMQ subscores showed that correlations are evident for Nonreactivity, Describing, Acting with Awareness, and Observing, weakest $r = .20$, highest $p = .04$, lowest $BF_{10} = 1.07$ (see Table 4 for details). No associations were found with affective empathy, apart from a negative correlation with Nonreactivity scores, $r = -.27$, $p = .01$, $BF_{10} = 5.71$.'

We have also revised the second paragraph of the Results section in a similar way (p. 12).

8. Discussion

This section of the paper was formatted in a different font from the rest of the paper. Please keep font consistent throughout the manuscript.

This has been corrected.

9. It is surprising that emotion recognition was not associated with empathy, as emotion recognition is widely recognized as one of the components of emotion regulation. Can the authors speculate further on why this study may not have found such an association?

We have added a new paragraph to the Discussion about this null result (p. 16-17):

'Empathy was associated with trait mindfulness, but not with emotion recognition performance. Bayesian analyses actually provided substantial-to-strong evidence for the null hypothesis. These results seem to contrast with those by Besel and Yuille (2010) showing an association between empathy and higher facial emotion recognition. However, this was found for fearful expressions only, and when participants were exposed to the stimuli for a long time. In other words, it could not be generalized across emotions and task conditions. More recently, Olderbak and Wilhelm (2017) conducted four studies with different designs and sample characteristics, and found that the relationship between empathy and facial emotion recognition is either not significant or weak. As for emotional prosody, to our knowledge no previous studies have documented associations with empathy, and studies focussing on potential correlations with personality traits have reported null results (Furnes, Berg, Mitchell, & Paulmann, 2019). It could be that emotion recognition and empathy reflect dissociable processes, with emotion recognition depending relatively more on low-level perceptual processes, and empathy on higher-order affective and cognitive mechanisms (e.g., Mitchell and Phillips, 2015). According to Olderbak and Wilhelm (2017), it could also be that the lack of an association reflects differences in the measurement approaches. While empathy is assessed as self-reported typical behaviour (i.e., a personality-like construct), emotion recognition is assessed as maximal effort using performance-based tasks.'

10. Despite presenting power analyses indicating that this study's sample size is sufficient to detect significant differences between variables, is it possible that sample size limitations contributed to the null findings? Is it possible that there was insufficient variability in the emotion recognition measures to detect differences across levels of mindfulness?

Sample size limitations is an unlikely explanation for our null findings: our sample of $N = 107$ was larger than the N indicated by the power analyses, and our Bayesian statistics provided formal evidence for the null hypothesis. This advantage of Bayesian statistics is now mentioned on p. 11.

The point about variability in the emotion recognition measures is a good one. We went back to our descriptive statistics, and there are indeed wide inter-individual differences, as indicated by participants' performance range and SD. These indices of variability compare well with research on the correlates of individual differences in emotion recognition, and therefore this seems an unlikely explanation for our findings. The following text has been added on p. 12:

'Despite the generally high performance, there was wide variability across participants, with scores ranging from $\cong .20$ to $\cong .88$ ($SD \cong .13$; Table 1). This is comparable to previous studies on the correlates of individual differences in emotion recognition (e.g., Correia et al., 2019; Israelashvili, Oosterwijk, Sauter, & Fischer, 2019; Scherer and Scherer, 2011).'

11. What are the limitations of this study?

We have identified two main limitations: our emotion recognition tasks do not allow us to assess emotion recognition in challenging conditions (e.g., under cognitive load), and this could have enhanced the likelihood of finding associations with mindfulness; and we only have a self-report measure of trait mindfulness, thus remaining unclear whether our results can be generalized to other types of measures. These limitations are discussed on p. 14-15:

'The potential role of task in associations between mindfulness and emotion recognition will need to be addressed in future studies. The benefits could be subtle and more likely to be seen in challenging conditions, which our tasks did not allow us to assess. This could be studied by systematically manipulating stimulus ambiguity or cognitive load (e.g.,

English et al., 2018; Lima et al., 2019). It will also be important to expand the current results, by combining measures of trait mindfulness like the one used here with other measures of mindfulness that do not rely on self-report. This could be done using, for example, experience-sampling methods or laboratory tasks of state mindfulness (e.g., Brown & Ryan, 2003).'

12. This paper would benefit from greater attention in both the Introduction and Discussion to the importance and implications of the authors' study question, as well as the findings of this study. Are there clinical implications for these findings, for example?

We have revised the Introduction and Discussion to emphasize the implications of our work, as suggested.

In the Introduction, we now describe the clinical benefits of mindfulness in more detail (p. 3), and the psychosocial and mental health correlates of emotion recognition skills (p. 5). New references have been added (Carton et al., 1999; Creswell, 2017). In the Discussion, the last paragraph has been extensively revised (p. 17):

'To conclude, the current study showed that trait mindfulness is not associated with the ability to recognize emotions in facial or prosodic expressions. We also found no associations between mindfulness and attention, but documented a robust link with cognitive empathy. Our results add to the growing literature on the correlates of trait mindfulness, thus contributing to a better understanding of this construct, and its role in positive psychological experience and skills (e.g., Bowlin & Baer, 2012; Brown & Ryan, 2003). They emphasize the notion that different aspects of emotional processing might relate to mindfulness in different ways. While trait mindfulness might serve important self-regulatory and awareness functions, its role for more low level emotional-perceptual mechanisms might be less apparent. These findings might also have broader implications for how mindfulness is used in clinical practice (e.g., for exploring the facets, potential benefits and limits of mindfulness with clients).'

Reviewer 2

1. This manuscript investigates the relation between mindfulness and the ability to recognize emotions expressed in the voice or face. I appreciate the comprehensive presentation of results, through the numerous tables, and that the data have been made open access. I also don't find the null results surprising, and in the area of socio-emotional research, I think null results are very informative.

The following are my concerns with this manuscript and/or suggestions for its improvement.

We thank the Reviewer for their careful reading of our work and for the constructive suggestions.

2. On page 4, please specify whether you are discussing trait emotional intelligence or ability emotional intelligence.

We are discussing trait emotional intelligence, and this is now mentioned (p. 4):

'Trait mindfulness has also been associated with higher trait emotional intelligence, as indicated by a meta-analysis of 17 studies (Miao, Humphrey, & Qian, 2018), and with attenuated emotional attentional blink (Makowski, Sperduti, Lavallée, Nicolas, & Piolino, 2019).'

3. In the introduction, but also for the lack of a relation between emotion perception and empathy, you might want to review the manuscript by Olderbak & Wilhelm (Emotion, 2017). They discuss how differences in measurement design,

namely typical behavior versus maximal effort, lowers correlations. That paper, because it presents results from 4 studies, is also a notable contrast to those who have found a positive relation between empathy and emotion perception.

Thank you for bringing our attention to this paper. We now cite it in the Introduction (p. 6), and review it in more detail in the Discussion (p. 16-17):

'Empathy was associated with trait mindfulness, but not with emotion recognition performance. Bayesian analyses actually provided substantial-to-strong evidence for the null hypothesis. These results seem to contrast with those by Besel and Yuille (2010) showing an association between empathy and higher facial emotion recognition. However, this was found for fearful expressions only, and when participants were exposed to the stimuli for a long time. In other words, it could not be generalized across emotions and task conditions. More recently, Olderbak and Wilhelm (2017) conducted four studies with different designs and sample characteristics, and found that the relationship between empathy and facial emotion recognition is either not significant or weak. As for emotional prosody, to our knowledge no previous studies have documented associations with empathy, and studies focussing on potential correlations with personality traits have reported null results (Furnes, Berg, Mitchell, & Paulmann, 2019). It could be that emotion recognition and empathy reflect dissociable processes, with emotion recognition depending relatively more on low-level perceptual processes, and empathy on higher-order affective and cognitive mechanisms (e.g., Mitchell and Phillips, 2015). According to Olderbak and Wilhelm (2017), it could also be that the lack of an association reflects differences in the measurement approaches. While empathy is assessed as self-reported typical behaviour (i.e., a personality-like construct), emotion recognition is assessed as maximal effort using performance-based tasks.'

4. Please provide more information about the sample. How were participants recruited? Was there inclusion criteria? Were there any dropouts or missing data? If so, how was that handled?

The recruitment method and inclusion criteria were added to the Participants section (p. 6):

'Participants were recruited in response to an advertisement (1) posted in local research participant pools, and (2) sent via-email to members of the wider community who were part of the researchers' social networks. Inclusion criteria were age ≥ 18 , no history of neurological or psychiatric disorders, normal hearing, normal or corrected-to-normal vision, and having European Portuguese as native language.'

There were no dropouts, but there were partial missing data from two participants, as explained on p. 10-11:

'Due to software malfunction, data from two participants on the ANT were not recorded. These participants were therefore excluded from all analyses including this variable.'

5. With the power analysis, please calculate what effect size could be identified, with the present sample and your power requirement.

Done. Thank you for the suggestion (p. 6-7):

'An *a priori* power analysis conducted with G*Power 3.1 (Faul, Erdfelder, Buchner, & Lang, 2009) indicated that a sample size of at least 84 was required to detect correlations of $r = .30$ or larger between variables, considering an alpha level of .05 and a power of .80. A sensitivity analysis, using the same alpha and power values, indicated that our actual sample ($N = 107$) had a sensitivity to detect correlations of at least $r = .27$.'

6. Please include cronbachs alpha for all measures, including all FFMQ subscales, and the ability tests.

Done (see p. 7, 8 and 10):

FFMQ:

'Internal consistency values were acceptable-to-excellent in the current dataset (Cronbach's $\alpha = .88$ for the full scale, ranging from $\alpha = .73$ for nonreactivity to $\alpha = .92$ for describing).'

QCAE:

'Internal consistency values were acceptable-to-good in the current dataset ($\alpha = .85$ for the full QCAE questionnaire, $\alpha = .79$ for affective empathy, and $\alpha = .87$ for cognitive empathy).'

Emotion recognition ability:

'Internal consistency values were acceptable-to-good for both tasks: $\alpha = .75$ for facial expressions, and $\alpha = .85$ for speech prosody.'

7. What was the name of the emotion perception tests?

These tests have no specific name. We programmed them ourselves, using previously validated stimuli and a standard response format, as described in detail on p. 9-10. Care was taken to avoid ceiling and floor effects, and to ensure that participants understood the tasks well, by completing practice trials before the experimental ones (p. 9).

8. Please explain why the bayesian analyses were conducted and what they add beyond the frequentist results.

Thank you for bringing this up, it was indeed unclear in our text.

The main reason why we combined frequentist and Bayesian statistics was that with Bayesian statistics we can quantify the evidence for the null hypothesis. We can formally draw inferences based on null results, which would be difficult based on p values. This was added on p. 11:

'A $BF_{10} < 1$ corresponds to evidence in favor of the null hypothesis: values between 0.33 and 1 correspond to anecdotal evidence, between 0.10 and 0.33 to substantial evidence, between 0.03 and 0.10 to strong evidence, between 0.01 and 0.03 to very strong evidence, and < 0.01 to decisive evidence. Thus, one important advantage of Bayesian statistics over the frequentist approach is that they allow us to interpret null results and to formally draw inferences based on them.'

9. Please provide a brief interpretation of the results, that accomodates the p value and BF10 value.

We now provide more details about the meaning of BF10 values on p. 11, and throughout the Results section we accommodate both p values and Bayesian statistics in our interpretation of findings (p. 12-13).

We also emphasize both types of analyses again in the Discussion (p. 13 and 16) and Abstract (p. 2).

P and BF10 values generally pointed to the same direction, that is, significant p values were associated with BF10 values indicating evidence for the alternative hypothesis. More relevant than this, however, when p values were non-significant, we could rely on BF10 values to formally draw inferences.

10. At the bottom of page 11, shouldn't it be BF_s10 instead of BF10? Or is the BF_s10 earlier a typo?

It should be BF10, not BF_s10, because we are referring to a single BF10 – the highest one of all BF_s10.

11. A path analysis in confirmatory factor analysis could be a better way to model the relations between mindfulness, empathy, attention, and emotion perception. That would also allow you to model latent variables, which allows one to relate constructs, instead of measures, thus controlling for measurement error.

We agree. Because most of our simple associations were non-significant, however (and BF10 values provided evidence for the null hypothesis), we thought that presenting more complex statistical models would make our results harder to read. We also note that we used one single measure for many of our constructs, in order to minimize measurement error, as the Reviewer rightly emphasizes. When we considered specific subscores, it was either as part of follow-up comparisons, or because those subscores reflect well-established constructs (e.g., cognitive and affective empathy).

Our initial analytic plan included mediation analyses (trait mindfulness → attention → emotion recognition; trait mindfulness → empathy → emotion recognition), which we have conducted, and as expected led to non-significant results (the 95% CI for indirect 'mediated' effects included 0 in both cases). Again, we decided not to report them because we thought they would not add to the paper, given the non-significant simple associations among most variables. We would be happy for this to be an Editorial decision, however.

12. Finally, in the discussion, comparing your null results with the significant findings of others, I wonder also about the statistical power in the other samples. Are your effect sizes comparable to what others have found? Were their measures more precise (i.e., higher internal consistency)?

This is very good point. We went back to the study by English et al. (2018), which to our knowledge is the only one that examined the relationship between trait mindfulness and facial emotional processing. They tested 126 participants (a sample size comparable to ours), and we now mention that in the Introduction (p. 5). They do not report effect sizes, but the internal consistency of their measures was similar to ours, and they have used the same trait mindfulness questionnaire as in the current study. We have added this information to the Discussion (p. 14):

'Our results also seem to contrast with those by English et al. (2018) pointing to a link between mindfulness and facial emotional processing. They have used the same mindfulness questionnaire as in the current study (FFMQ), and the internal consistency of their self-report and performance measures is comparable to ours. The discrepancy might stem from differences in samples, or the particular way emotion recognition was assessed. English et al. (2018) asked participants to recognize emotions in ten-image sequences starting with a neutral expression and ending with a full expression. Their focus was on the *amount* of information needed for recognition, not on the ability to recognize emotions. Moreover, their results were mostly seen for fearful expressions only, as assessed exclusively by female participants.'

We also emphasize that the null correlation between mindfulness and attention performance can be seen in previous studies, even if they are well-powered and include reliable measures (p. 15).

We thank again both Reviewers for their careful reading of our work and for the constructive comments.